# An Attribution Method for Siamese Encoders

**Lucas Möller** and **Dmitry Nikolaev** and **Sebastian Padó**
Institute for Natural Language Processing, University of Stuttgart, Germany
{lucas.moeller, dmitry.nikolaev, pado}@ims.uni-stuttgart.de

## Abstract

Despite the success of Siamese encoder models such as sentence transformers (ST), little is known about the aspects of inputs they pay attention to. A barrier is that their predictions cannot be attributed to individual features, as they compare two inputs rather than processing a single one. This paper derives a local attribution method for Siamese encoders by generalizing the principle of integrated gradients to models with multiple inputs. The output takes the form of feature-pair attributions and in case of STs it can be reduced to a token–token matrix. Our method involves the introduction of *integrated Jacobians* and inherits the advantageous formal properties of integrated gradients: it accounts for the model's full computation graph and is guaranteed to converge to the actual prediction. A pilot study shows that in case of STs few token pairs can dominate predictions and that STs preferentially focus on nouns and verbs. For accurate predictions, however, they need to attend to the majority of tokens and parts of speech.

## 1 Introduction

Siamese encoder models (SE) process two inputs concurrently and map them onto a single scalar output. One realization are sentence transformers (ST), which learn to predict a similarity judgment between two texts. They have lead to remarkable improvements in many areas including sentence classification and semantic similarity (Reimers and Gurevych, 2019), information retrieval (IR) (Thakur et al., 2021) and automated grading (Bexte et al., 2022). However, little is known about aspects of inputs that these models base their decisions on, which limits our understanding of their capabilities and limits.

Nikolaev and Padó (2023) analyze STs with sentences of pre-defined lexical and syntactic structure and use regression analysis to determine the relative importance of different text properties. MacAvaney et al. (2022) analyze IR models with samples consisting of queries and contrastive documents that differ in certain aspects. Opitz and Frank (2022) train an ST to explicitly encode AMR-based properties in its sub-embeddings.

More is known about the behavior of standard transformer models; see Rogers et al. (2020) for an overview. Hidden representations have been probed for syntactic and semantic information (Tenney et al., 2019; Conia and Navigli, 2022; Jawahar et al., 2019). Attention weights have been analyzed with regard to linguistic patterns they capture (Clark et al., 2019; Voita et al., 2019) and have been linked to individual predictions (Abnar and Zuidema, 2020; Vig, 2019). However, attention weights alone cannot serve as explanations for predictions (Jain and Wallace, 2019; Wiegreffe and Pinter, 2019). To obtain *local* explanations for individual predictions (Li et al., 2016), Bastings and Filippova (2020) suggest the use of feature attribution methods (Danilevsky et al., 2020). Among them, *integrated gradients* are arguably the best choice due to their strong theoretic foundation (Sundararajan et al., 2017; Atanasova et al., 2020) (see Appendix A). However, such methods are not directly applicable to Siamese models, which compare two inputs instead of processing a single one.

In this work, we derive attributions for an SE's predictions to its inputs. The result takes the form of pair-wise attributions to features from the two inputs. For the case of STs it can be reduced to a token–token matrix (Fig. 1). Our method takes into account the model's full computational graph and only requires it to be differentiable. The combined prediction of all attributions is theoretically guaranteed to converge against the actual prediction. To the best of our knowledge, we propose the first method that can accurately attribute predictions of Siamese models to input features. Our code is publicly available.[1]

---

[1] https://github.com/lucasmllr/xsbert

## 2 Method

### 2.1 Feature-Pair Attributions

Let $f$ be a Siamese model with an encoder $\mathbf{e}$ which maps two inputs $\mathbf{a}$ and $\mathbf{b}$ to a scalar score $s$:

$$f(\mathbf{a}, \mathbf{b}) = \mathbf{e}^T(\mathbf{a})\,\mathbf{e}(\mathbf{b}) = s \qquad (1)$$

Additionally, let $\mathbf{r}$ be *reference inputs* that always result in a score of zero for any other input $\mathbf{c}$: $f(\mathbf{r}, \mathbf{c}) = 0$. We extend the principle that Sundararajan et al. (2017) introduced for single-input models (Appendix A) to the following ansatz for two-input models, and reformulate it as an integral:

$$f(\mathbf{a}, \mathbf{b}) - f(\mathbf{a}, \mathbf{r}_a) - f(\mathbf{b}, \mathbf{r}_b) + f(\mathbf{r}_a, \mathbf{r}_b)$$

$$= \int_{\mathbf{r}_b}^{\mathbf{b}} \int_{\mathbf{r}_a}^{\mathbf{a}} \frac{\partial^2}{\partial x_i \partial y_j} f(\mathbf{x}, \mathbf{y})\; dx_i\, dy_j \qquad (2)$$

$$= \sum_{ij} (\mathbf{a} - \mathbf{r}_a)_i \left(\mathbf{J}_a^T \mathbf{J}_b\right)_{ij} (\mathbf{b} - \mathbf{r}_b)_j$$

This Ansatz is entirely general to any model with two inputs. In the last line, we then make explicit use of the Siamese architecture to derive the final attributions (details in Appendix B). Indices $i$ and $j$ are for dimensions of the two inputs $\mathbf{a}$ and $\mathbf{b}$, respectively. Individual summands on the right-hand-side can be expressed in an attribution matrix, which we will refer to as $\mathbf{A}_{ij}$.

By construction, all terms involving a reference input on the left-hand-side vanish, and the sum over this attribution matrix is exactly equal to the model prediction:

$$f(\mathbf{a}, \mathbf{b}) = \sum_{ij} \mathbf{A}_{ij}(\mathbf{a}, \mathbf{b}) \qquad (3)$$

In the above result, we define the matrices $\mathbf{J}$ as:

$$(\mathbf{J}_a)_{ki} = \int_{\alpha=0}^{1} \frac{\partial \mathbf{e}_k(\mathbf{x}(\alpha))}{\partial \mathbf{x}_i}\, d\alpha$$
$$\approx \frac{1}{N} \sum_{n=1}^{N} \frac{\partial \mathbf{e}_k(\mathbf{x}(\alpha_n))}{\partial \mathbf{x}_i} \qquad (4)$$

The expression inside the integral, $\partial \mathbf{e}_k / \partial \mathbf{x}_i$, is the Jacobian of the encoder, i.e. the matrix of partial derivatives of all embedding components $k$ w.r.t. all input components $i$. We therefore, call $\mathbf{J}$ an *integrated Jacobian*. The integral proceeds along positions $\alpha$ on an integration path formed by the linear interpolation between the reference $\mathbf{r}_a$ and input $\mathbf{a}$: $\mathbf{x}(\alpha) = \mathbf{r}_a + \alpha(\mathbf{x} - \mathbf{r}_a)$.

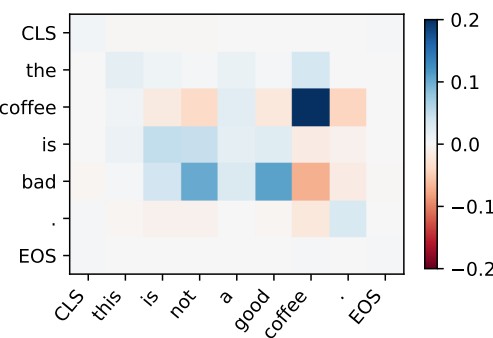

Figure 1: An example token–token attribution matrix to layer nine. The model correctly relates *not... good* to *bad* and matches *coffee*. Similarity score: 0.82, attribution error: $10^{-3}$ for $N = 500$.

Intuitively, Eq. 4 embeds all inputs between $\mathbf{r}_a$ and $\mathbf{a}$ along the path $\mathbf{x}(\alpha)$ and computes their sensitivities w.r.t. input dimensions (Samek et al., 2017). It then collects all results on the path and combines them into the matrix $\mathbf{J}_\mathbf{a}$; analogously for $\mathbf{J}_b$. Eq. 2 combines the sensitivities of both inputs and computes pairwise attributions between all feature combinations in $\mathbf{a}$ and $\mathbf{b}$.

In a transformer model, text representations are typically of shape $S \times D$, where $S$ is the sequence length and $D$ is the embedding dimensionality. Therefore, $\mathbf{A}$ quickly becomes intractably large. Fortunately, the sum in Eq. 2 allows us to combine individual attributions. Summing over the embedding dimension $D$ yields a matrix of shape $S_\mathbf{a} \times S_\mathbf{b}$, the lengths of the two input sequences. Figure 1 shows an example.

Since Eq. 3 is an equality, the attributions provided by $\mathbf{A}$ are provably correct and we can say that they *faithfully* explain which aspects of the inputs the model regards as important for a given prediction. For efficient numerical calculation, we approximate the integral by a sum of $N$ steps corresponding to equally spaced points $\alpha_n$ along the integration path (Eq. 4). The resulting approximation error is guaranteed to converge to zero as the sum converges against the integral. It is further perfectly quantifiable by taking the difference between the left- and right-hand side in Eq. 3 (cf. § 3.2).

### 2.2 Adapting Existing Models

For our attributions to take the form of Eq. 3, we need to adapt standard models in two aspects:

**Reference input.** It is crucial that $f$ consistently yields a score of zero for inputs involving a reference $\mathbf{r}$. A solution would be to set $\mathbf{r}$ to an input

that the encoder maps onto the zero vector, so that $f(\mathbf{c}, \mathbf{r}) = e^T(\mathbf{c})\, e(\mathbf{r}) = e^T(\mathbf{c})\, \mathbf{0} = 0$. However, it is not trivial to find such an input. We avoid this issue by choosing an arbitrary reference and shifting all embeddings by $\mathbf{r}$ in the embedding space, $e(\mathbf{c}) = e'(\mathbf{c}) - e'(\mathbf{r})$, where $e'$ is the original encoder, so $e(\mathbf{r}) = \mathbf{0}$. For simplicity, we use a sequence of padding tokens with the same length as the respective input as reference $\mathbf{r}$.

**Similarity measure.**   Sentence transformers typically use cosine distance to compare embeddings, normalizing them to unit length. Unfortunately, normalization of the zero vector, which we map the reference to, is undefined. Therefore, we replace cosine distance with the (unnormalized) dot product when computing scores as shown in Eq. 1.

## 2.3   Intermediate Representations

Different from other deep models, in transformers, due to the sequence-to-sequence architecture and the language-modeling pre-training, intermediate representations still correspond to (the contexts of) input tokens. Therefore, attributing predictions to inputs is one option, but it is also interesting to consider attributions to intermediate and even output representations. In these cases, $f$ maps the given intermediate representation to the output. Attributions then explain, which dimensions within this representation the model consults for its prediction.

## 3   Experiments and Results

In our experiments, we evaluate the predictive performance of different model configurations and then test their attribution accuracy. Generally, the two are independent, so that a model with excellent attribution ability may not yield excellent predictions or vice versa. In the following, we analyze statistical characteristics of attributions. To demonstrate our method, we perform a pilot on which parts of speech (POS) models attend to.

### 3.1   Predictive Performance

We begin by evaluating how much the shift of embeddings and the change of objective affect the predictive performance of STs. To this end, we fine-tune STs off different pre-trained base models on the widely used semantic text similarity (STS) benchmark (Cer et al., 2017) We tune all base models in two different configurations: the standard setting for Siamese sentence transformers (NON-ADJUSTED, Reimers and Gurevych 2019), and with

| Base model | adjusted | cosine | dot |
|---|---|---|---|
| S-MPNet | ✓ | **85.9** | **82.6** |
| | ✗ | 87.6 | 83.9 |
| S-distillRoBERTa | ✓ | 85.7 | 80.7 |
| | ✗ | 86.3 | 77.4 |
| MPNet | ✓ | 85.1 | 80.4 |
| | ✗ | 86.3 | 84.2 |
| distillRoBERTa | ✓ | 80.4 | 73.4 |
| | ✗ | 84.6 | 76.2 |
| RoBERTa | ✓ | 77.7 | 68.8 |
| | ✗ | 86.1 | 68.8 |

Table 1: Spearman correlations between labels and scores computed by cosine distance and dot product of embeddings. We evaluate pre-trained sentence transformers (top) and vanilla transformers (bottom). *Adjusted* indicates modification according to Sec. 2.2. Best results for (non-)adjusted models are (underlined) bold.

our adjustments from § 2.2 applied for the model to obtain exact-attribution ability (ADJUSTED). Training details are provided in Appendix H. For all models, we report Spearman correlations between predictions and labels for both cosine distance and dot product of embeddings.

Our main focus is on already pre-trained sentence transformers. Results for them are shown in the top half of Table 1. Generally, adjusted models cannot reach the predictive performance of standard STs. However, the best adjusted model (S-MPNet) only performs 1.7 points worse (cosine) than its standard counterpart. This shows that the necessary adjustments to the model incur only a modest price in terms of downstream performance.

The bottom half of the table shows performances for vanilla transformers that have only been pre-trained on language modeling tasks. Results for these models are more diverse. However, we do not expect their predictions to be comparable to STs, and we mostly include them to evaluate attribution accuracies on a wider range of models below.

### 3.2   Attribution Accuracy

As shown in § 2.1, all attributions in $\mathbf{A}$ must sum up to the predicted score $s$ if the two integrated Jacobians are approximated well by the sum in Eq. 4. We test how many approximation steps $N$ are required in practice and compute the absolute error between the sum of attributions and the prediction score as a function of $N$ for different intermediate representations. Fig. 2 shows the results for the

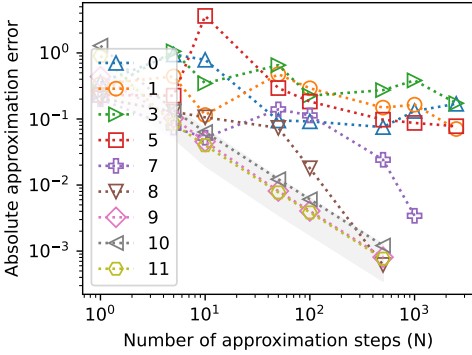

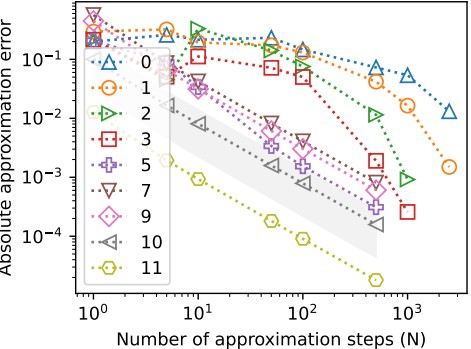

Figure 2: Layer-wise attribution errors for the S-MPNet (top) and the RoBERTa based model (bottom). Standard deviations are shown exemplary.

S-MPNet model. Generally, attributions to deeper representations, which are closer to the output, can be approximated with fewer steps. Attributions to e.g. layer 9 are only off by $(5\pm5)\times10^{-3}$ with as few as $N=50$ approximation steps. Layer 7 requires $N=1000$ steps to reach an error of $(2\pm3)\times10^{-3}$ and errors for shallower layers have not yet started converging for as many as $N=2500$ steps, in this model. In contrast, in the equally deep RoBERTa model, errors for attributions to all layers including input representations have started to converge at this point. The error for attributions to input representations remains at only $(1\pm1)\times10^{-2}$ – evidently, attribution errors are highly model specific.

Our current implementation and resources limit us to $N \leq 2500$. However, we emphasize that this is not a fundamental limit. The sum in Equation 4 converges against the integral for large $N$, thus it is only a matter of computational power to achieve accurate attributions to shallow layers in any model.

### 3.3 Distribution of Attributions

For an overview of the range of attributions that our best-performing model S-MPNet assigns to pairs of tokens, Fig. 3 shows a histogram of attri-

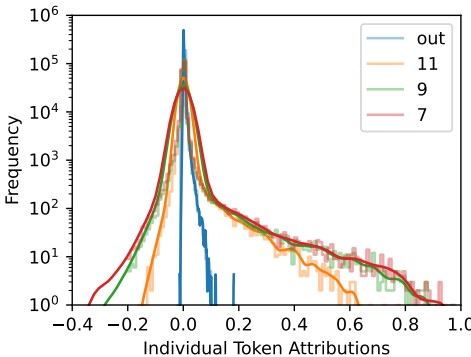

Figure 3: Distribution of individual token–token attributions to different intermediate representations of the S-MPNet model.

butions to different (intermediate) representations across 1000 STS test examples. A large fraction of all attributions to intermediate representations is negative (38% for layer 11). Thus, the model can balance matches and mismatches. This becomes apparent in the example in Fig. 4. The word *poorly* negates the meaning of the sentence and contributes negatively to the prediction. Interestingly, attributions to the output representation do not capture this characteristic, as they are almost exclusively positive (95%). Other models behave similarly (Appendix E).

It further interests us how many feature-pairs the model typically takes into consideration for individual predictions. We sort attributions by their absolute value and add them up cumulatively. Averaging over 1000 test-instances results in Fig. 5. The top 5% of attributions already sum up to $(77\pm133)\%$ [2] of the model prediction. However, the large standard deviation (blue shading in Fig. 5) shows that these top attributions alone do not yet reliably explain predictions for all sentence pairs. For a trustworthy prediction with a standard deviation below 5% (2%), the model requires at least 78% (92%) of all feature-pairs.

### 3.4 POS Relations

We evaluate which combinations of POS the model relies on to compute similarities between sentences. For this purpose, we combine token- to word-attributions by averaging. We then tag words with a POS-Classifier.[3]

Fig. 6 shows shares of the ten most frequent POS-relations among the highest 10%, 25%, and 50%

---

[2]cumulative sums of top attributions can be negative.
[3]https://huggingface.co/flair/pos-english

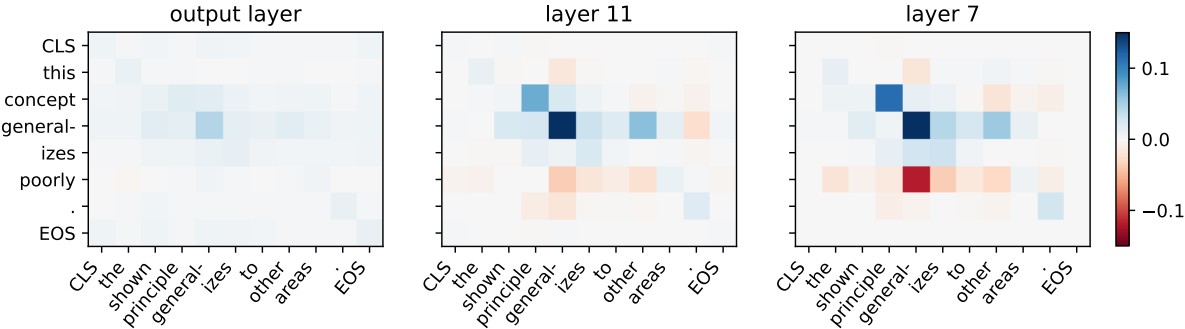

Figure 4: Attributions of the same example to different representations in the S-MPNet model.

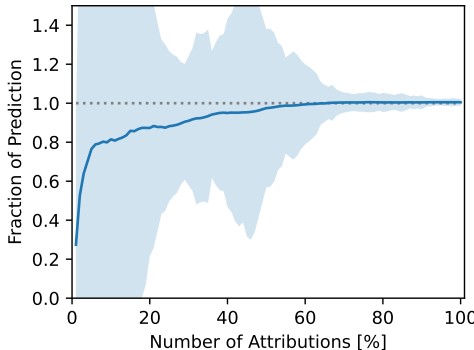

Figure 5: Mean cumulative prediction and standard-deviation of token–token attributions sorted by their absolute value.

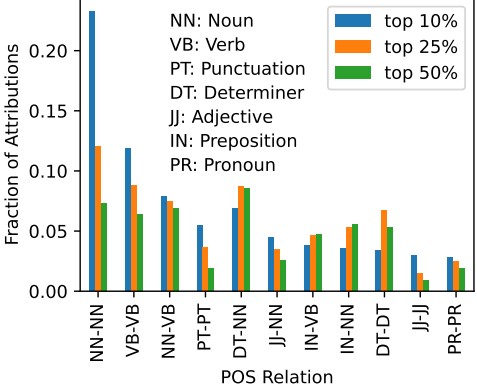

Figure 6: Distribution of the highest 10%, 25% and 50% attributions among the most attributed parts of speech.

of attributions on the STS test set. Within the top 10%, noun-noun attributions clearly dominate with a share of almost 25%, followed by verb-verb and noun-verb attributions. Among the top 25% this trend is mitigated, the top half splits more evenly. When we compute predictions exclusively from attributions to specific POS-relations, nouns and verbs together explain $(53 \pm 90)\%$, and the top ten POS-relations (cf. Fig. 6) account for $(66 \pm 98)\%$ of the model prediction. The 90% most important relations achieve $(95 \pm 29)\%$. Thus, the model largely relies on nouns (and verbs) for its predictions. This extends the analysis of Nikolaev and Padó (2023), who find in a study on synthetic data that SBERT similarity is determined primarily by the lexical identities of arguments (subjects / objects) and predicates of matrix clauses. Our findings show that this picture extends largely to naturalistic data, but that it is ultimately too simplistic: on the STS corpus, the model does look beyond nouns and verbs, taking other parts of speech into account to make predictions.

## 4 Conclusion

Our method can provably and accurately attribute Siamese model predictions to input and intermediate feature-pairs. While in sentence transformers output attributions are not very expressive and attributing to inputs can be computationally expensive, attributions to deeper intermediate representations are efficient to compute and provide rich insights.

Referring to the terminology introduced by Doshi-Velez and Kim (2017) our feature-pair attributions are single *cognitive chunks* that combine additively in the model prediction. Importantly, they can explain which feature-pairs are relevant to individual predictions, but not why (Lipton, 2018).

Improvements may be achieved by incorporating the discretization method of Sanyal and Ren (2021), and care must be applied regarding the possibility of adversarially misleading gradients (Wang et al., 2020). In the future, we believe our method can serve as a diagnostic tool to better analyze the predictions of Siamese models.

## Limitations

The most important limitation of our method is the fact that the original model needs to be adjusted and fine-tuned in order to adopt to the shift of embeddings and change of objective that we introduced in Section 2.2. This step is required because the dot-product (and cosine-similarity) of shifted embeddings does not equal that of the original ones.[4] Therefore, we cannot directly analyze off-the-shelf models.

Second, when a dot-product is used to compare two embeddings instead of a cosine-distance, self-similarity is not preserved: without normalization, the dot-product of an embedding vector with itself is not necessarily one.

Third, our evaluation of predictive performance is limited to the task of semantic similarity and the STS benchmark (which includes multiple datasets). This has two reasons: we focus on the derivation of an attribution method for Siamese models and the evaluation of the resulting attributions. The preservation of embedding quality for downstream tasks in non-Siamese settings is out of the scope of this short paper.

## Ethics Statement

Our work does not involve sensitive data nor applications. Both, the used pre-trained models and datasets are publicly available. Computational costs for the required fine-tuning are relatively cheap. We believe our method can make Siamese models more transparent and help identify potential errors and biases in their predictions.

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

## A  Integrated Gradients

Our method builds on the principle that was introduced by Sundararajan et al. (2017) for models with a single input. Here we derive the core concept of their *integrated gradients*.

Let $f$ be a differentiable model taking a single vector valued input $\mathbf{x}$ and producing a scalar output $s \in [0, 1]$: $f(\mathbf{x}) = s$. In addition let $\mathbf{r}$ be a *reference input* yielding a neutral output: $f(\mathbf{r}) = 0$. We can then start from the difference in the two inputs and reformulate it as an integral (regarding $f$ an anti-derivative):

$$f(\mathbf{a}) - f(\mathbf{r}) = \int_{\mathbf{r}}^{\mathbf{a}} \frac{\partial f(\mathbf{x})}{\partial \mathbf{x}_i} d\mathbf{x}_i \qquad (5)$$

This is a path integral from the point $\mathbf{r}$ to $\mathbf{a}$ in the input space. We use component-wise notation, and double indices are summed over. To solve the integral, we parameterize the path from $\mathbf{r}$ to $\mathbf{a}$ by the straight line $\mathbf{x}(\alpha) = \mathbf{r} + \alpha(\mathbf{a} - \mathbf{r})$ and substitute it:

$$= \int_{\alpha=0}^{1} \frac{\partial f(\mathbf{x}(\alpha))}{\partial \mathbf{x}_i(\alpha)} \frac{\partial \mathbf{x}_i(\alpha)}{\partial \alpha} d\alpha \qquad (6)$$

The first term inside the above integral is the gradient of $f$ at the position $\mathbf{x}(\alpha)$. The second term is the derivative of the straight line and reduces to $d\mathbf{x}(\alpha)/d\alpha = (\mathbf{a} - \mathbf{r})$, which is independent of $\alpha$ and can be pulled out of the integral:

$$= (\mathbf{a} - \mathbf{r})_i \int_{\alpha=1}^{1} \nabla_i f(\mathbf{x}(\alpha)) \, d\alpha \qquad (7)$$

This last expression is the contribution of the $i^{th}$ input feature to the difference in Equation 5. If $f(\mathbf{r}) = 0$, then the sum over all contributions equals the model prediction $f(\mathbf{a}) = s$. Note, that the equality between Equation 5 and Equation 7 holds strictly. Therefore, Equation 7 is an exact reformulation of the model prediction.

## B   Detailed Derivation

For the case of a model receiving two inputs, we extend the ansatz from Equation 5 to:

$$f(\mathbf{a}, \mathbf{b}) - f(\mathbf{a}, \mathbf{r}_b) - f(\mathbf{b}, \mathbf{r}_b) + f(\mathbf{r}_a, \mathbf{r}_b)$$

$$= \big[ f(\mathbf{a}, \mathbf{b}) - f(\mathbf{r}_a, \mathbf{b}) \big] - \big[ f(\mathbf{a}, \mathbf{r}_b) - f(\mathbf{r}_a, \mathbf{r}_b) \big]$$

$$= \int_{\mathbf{r}_b}^{\mathbf{b}} \frac{\partial}{\partial \mathbf{y}_j} \big[ f(\mathbf{a}, \mathbf{y}) - f(\mathbf{r}_a, \mathbf{y}) \big] \, d\mathbf{y}_j$$

$$= \int_{\mathbf{r}_b}^{\mathbf{b}} \int_{\mathbf{r}_a}^{\mathbf{a}} \frac{\partial^2}{\partial \mathbf{x}_i \partial \mathbf{y}_j} f(\mathbf{x}, \mathbf{y}) \, d\mathbf{x}_i \, d\mathbf{y}_j$$

$$\qquad (8)$$

We plug in the definition of the Siamese model (Equation 1), using element-wise notation for the output embedding dimensions $k$, and again, omit sums over double indices:

$$= \int_{\mathbf{r}_a}^{\mathbf{a}} \int_{\mathbf{r}_b}^{\mathbf{b}} \frac{\partial^2}{\partial \mathbf{x}_i \partial \mathbf{y}_j} \mathbf{e}_k(\mathbf{x}) \, \mathbf{e}_k(\mathbf{y}) \, d\mathbf{x}_i \, d\mathbf{y}_j \qquad (9)$$

Neither encoding depends on the other integration variable, and we can separate derivatives and integrals:

$$= \int_{\mathbf{r}_a}^{\mathbf{a}} \frac{\partial \mathbf{e}_k(\mathbf{x})}{\partial \mathbf{x}_i} d\mathbf{x}_i \int_{\mathbf{r}_b}^{\mathbf{b}} \frac{\partial \mathbf{e}_k(\mathbf{y})}{\partial \mathbf{y}_j} d\mathbf{y}_j \qquad (10)$$

Different from above, the encoder $e$ is a vector-valued function. Therefore, $\partial \mathbf{e}_k(\mathbf{x})/\partial \mathbf{x}_i$ is a Jacobian, not a gradient. We integrate along straight lines from $\mathbf{r}$ to $\mathbf{a}$, and from $\mathbf{r}_a$ to $\mathbf{b}$, parameterized by $\alpha$ and $\beta$, respectively, and receive:

$$= (\mathbf{a} - \mathbf{r}_a)_i \left[ \int_\alpha \frac{\partial \mathbf{e}_k(\mathbf{x}(\alpha))}{\partial \mathbf{x}_i} d\alpha \right. $$
$$\left. \int_\beta \frac{\partial \mathbf{e}_k(\mathbf{y}(\beta))}{\partial \mathbf{y}_j} d\beta \right] (\mathbf{b} - \mathbf{r}_b)_j$$
$$\qquad (11)$$

With the definition of *integrated Jacobians* from Equation 4, we can use vector notation and write the sum over the output dimension $k$ in square brackets as a matrix product: $J_\mathbf{a}^T J_\mathbf{b}$. If $\mathbf{r}$ consistently yields a prediction of zero, the last three terms on the left-hand-side of Equation 8 vanish, and we arrive at our result in Equation 2, where we denote the sum over input dimensions $i$ and $j$ explicitly.

## C   Intermediate Attributions

Fig. 4 shows attributions for one example to different representations in the S-MPNet model. Attributions to layer eleven and seven capture the negative contribution of *poorly*, which is completely absent in the output layer attributions. As Fig. 3 shows output attributions are less pronounced and almost exclusively positive.

## D   Attribution Accuracy

In Fig. 7 we include the attribution accuracy plot for the shallower S-distillRoBERTa model. Attributions to all layers converge readily for small $N$.

## E   Attribution Distribution

Fig. 8 shows distribution plots for attributions to different intermediate representations of the RoBERTa and the S-distillRoBERTa models. In both cases we also observe positivity of attributions to the output representation. For RoBERTa this characteristic proceeds to the last encoder layers.

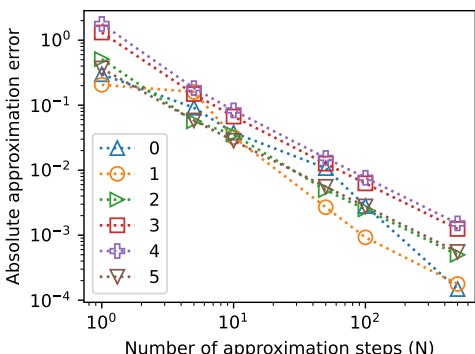

Figure 7: Layer-wise attribution errors for the distilled Roberta based model

## F  Different Models

Attributions of different models can characterize differently even if agreement on the overall score is good. Fig. 9 shows two examples.

## G  Prediction Failures

Fig. 10 shows examples in which the S-MPNet prediction is far off from the label. In the future, a systematic analysis of such cases could provide insights into where the model fails.

## H  Training Details

We fine-tune all models in a Siamese setting on the STS-benchmark train split. Models either use shifted embeddings combined with a dot-product objective or normal embeddings together with a cosine objective. All trainings run for five epochs, with a batch size of 16, a learning rate of $2 \times 10^{-5}$ and a weight decay of 0.1 using the AdamW-optimizer. 10% of the training data is used for linear warm-up

## I  Implementation

This sections intends to bridge the gap between the shown theory and its implementation. In Eq. 4 $\mathbf{e}(\mathbf{x}(\alpha_n))$ is a single forward pass for the input $\mathbf{x}(\alpha_n)$ through the encoder $\mathbf{e}$. $\partial \mathbf{e}_k(\mathbf{x}(\alpha_n))/\partial \mathbf{x}_i$ is the corresponding backward pass of the $k^{th}$ embedding dimension w.r.t. the $i^{th}$ input (or intermediate) dimension. In order to calculate either *integrated Jacobian*, $N$ such passes through the model need to be computed for all interpolation steps $n \in \{1,...,N\}$ along the integration paths between references and inputs.

Fortunately, they are independent for different interpolation steps and we can batch them for parallel

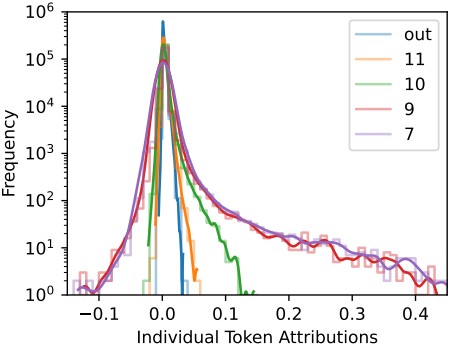

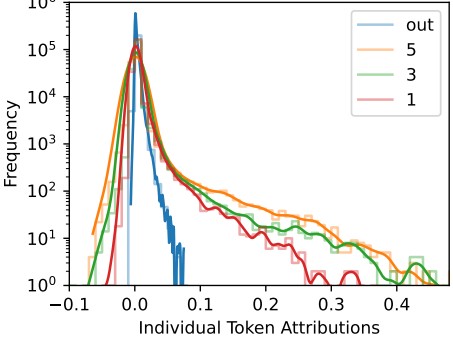

Figure 8: Attribution Distributions for the RoBERTa-based model (top), and the S-distillRoBERTa model (bottom).

| Model | Link |
|---|---|
| S-MPNet | all-mpnet-base-v2 |
| S-distillRoBERTa | all-distilroberta-v1 |
| MPNet | mpnet-base |
| distillRoBERTa | distilroberta-base |
| RoBERTa | roberta-base |

Table 2: Links to huggingface weights of the used models.

computation. Regarding computational complexity, this process hence requires $N/B$ forward and backward passes through the encoder, where $B$ is the used batch size. Attributions to intermediate representations do not require the full backward pass and are thus computationally cheaper. Once the two *integrated Jacobians* are derived, the computation of the final attribution matrix in the last line of Eq. 8 is a matter of matrix multiplication.

## J  Model Weights

Table 2 includes links to the huggingface model weights that we use in this paper.

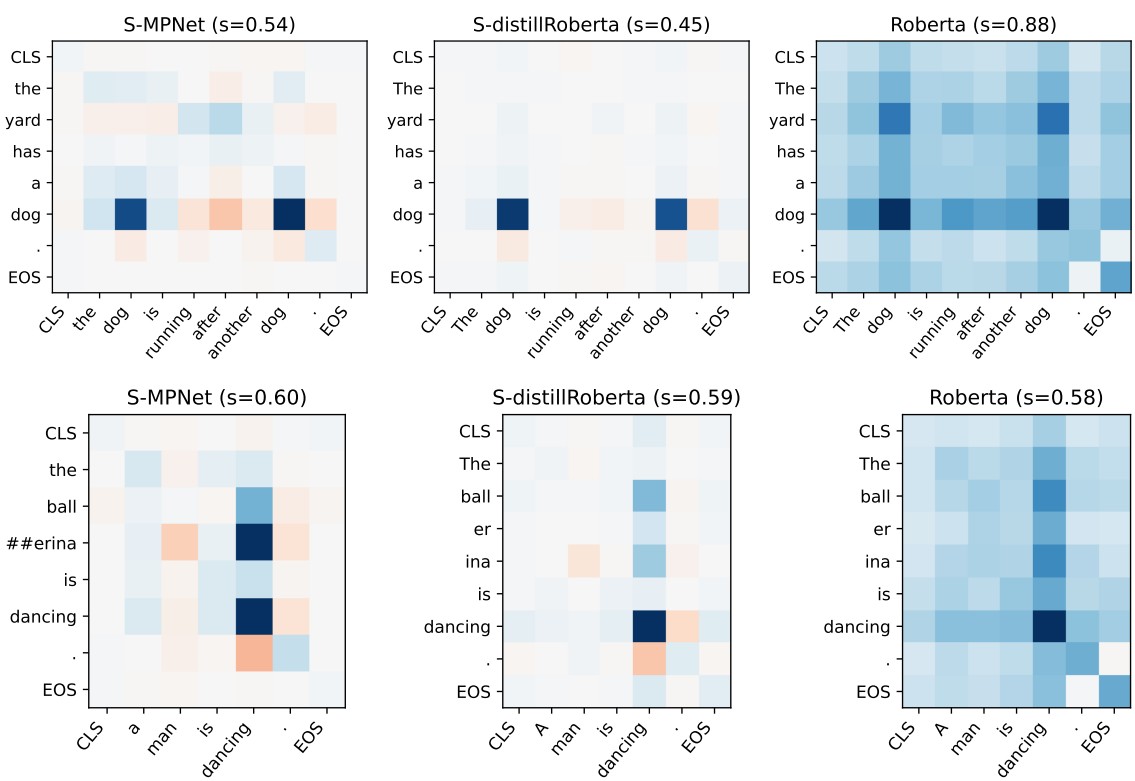

Figure 9: Attributions for identical sentences by different models. Model and scores are given in the titles.

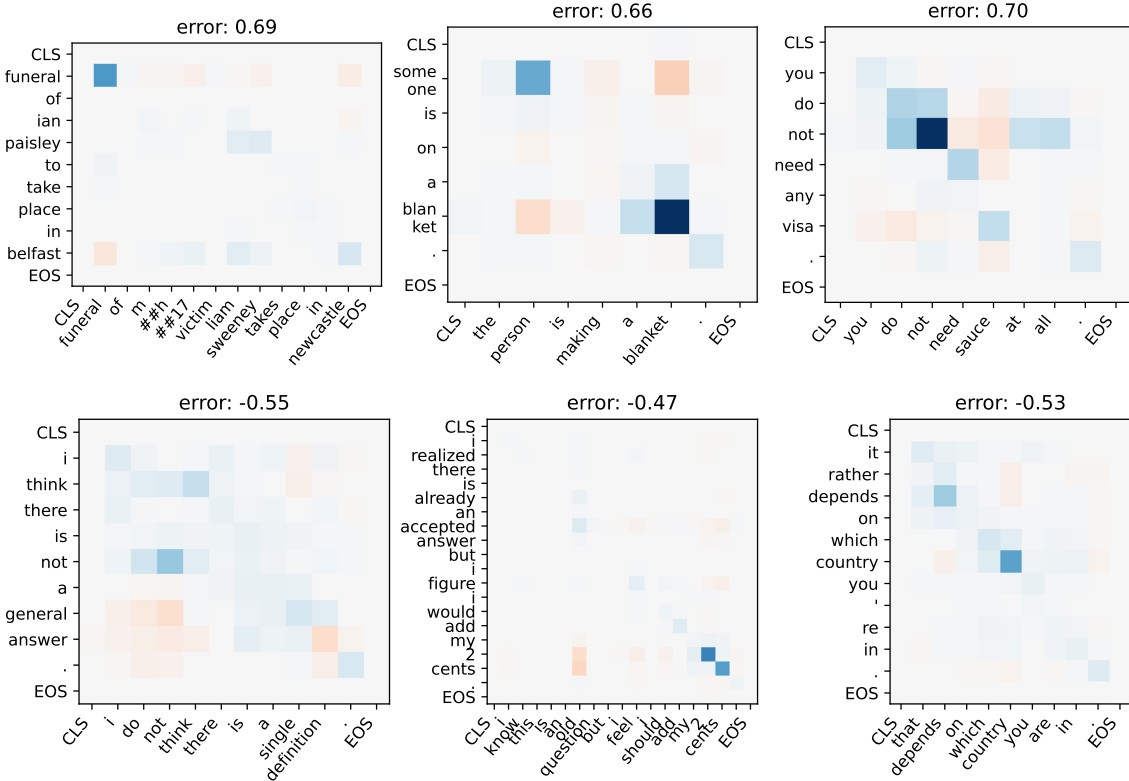

Figure 10: Failure cases of the M-PNet. Examples in the top row show over estimations, the bottom row shows under estimations of semantic similarity.