# OpenReview forum: "An Attribution Method for Siamese Encoders"
_EMNLP/2023/Conference — EMNLP 2023 Main_

### Official Review · Reviewer_Ug1E · 2023-07-24

**Soundness:** 4

**Excitement:**

4: Strong: This paper deepens the understanding of some phenomenon or lowers the barriers to an existing research direction.

**Justification For Ethical Concerns:**

As the authors noted, I don't believe the proposed method introduce additional ethical concerns than the ones originating from benchmark datasets.

**Paper Topic And Main Contributions:**

The work is about understanding which parts of the sentence representations attribute to their classification results, in the Siamese encoder setting (especially in the Siamese transformer (ST) setting).
The motivation of this work comes from the observation that unlike cross-sentence attention models which directly compare two sentences as part of model components, in the Siamese setting it is hard to figure out pair-wise attributions of two inputs.
The works starts from extending the attribution method designed for single-input models (Sundararajan et al. 2017), where a vector which outputs zero-score is used as a reference for measure attributions.
Specifically, the authors rewrite the score function (dot production) with the two displacement vectors and integrated Jacobians.
By switching the scoring function from cosine similarity to dot product, to make a reference vector exist, the derivation of the scoring function can be derive expressed with the above terms can be done exactly, up to approximation error introduced by summation instead of exact integration.
From experiments on multiple models and their layer outputs, it is empirically proven that the proposed method exhibits the desired effect, by responding to identical or similar words, or negation words.


**Questions For The Authors:**

- The method requires evaluating on multiple points along with the line between a, b and r. How will be the computational overhead of applying this method?

**Reasons To Accept:**

- The extension of Sundararajan et al. (2017)'s method into Siamese setting is done naturally and thoroughly
- Various analyses comparing multiple layers and properties are given.

**Reasons To Reject:**

- From my understanding, the method can only be applied to sequence-to-sequence sentence encoder setting, i.e. we have multiple output vectors in token-level
- More elaboration on experiment settings will be thankful
  - For me, it's not clear what 1.7 points of sacrifice (line 175) means.

**Reproducibility:**

3: Could reproduce the results with some difficulty. The settings of parameters are underspecified or subjectively determined; the training/evaluation data are not widely available.

**Reviewer Confidence:**

3: Pretty sure, but there's a chance I missed something. Although I have a good feel for this area in general, I did not carefully check the paper's details, e.g., the math, experimental design, or novelty.

---

> ### Author Rebuttal · Authors · 2023-08-28
>
> Thank you for your feedback. In the following we reply to your main concerns.
>
> ### Restriction to Seq-to-Seq models
> Our method is applicable to any kind of Siamese model and is not restricted to sequence-to-sequence encoders. In the detailed derivation (Appendix B) this is mirrored in the fact that we only require an abstract formulation of an encoder $e$ and do not need to make any further assumptions about this model component. If $e$ was not a token-based seq-to-seq model the result would be a feature-feature matrix and could not be reduced to a token-token version, but the method would still be applicable as it is. Presumably, a different kind of aggregation by summation over the entries of the attribution matrix could be carried out (cf. lines 118-126).
> An interesting aspect of transformer models is the fact that all intermediate representations are of the same shape and can be associated with individual input tokens (Section 2.3). We make direct use of this property. However, our method does not depend on it as it can compute attributions to any representation — input or intermediate ones.
>
> ### Clarity and detail
> We agree that Section 3.1 together with Table 1 are currently not reader friendly and acknowledge that all reviewers criticize this part. We will update it in the revised version.
>
> ### Computational complexity
> Generally, our method requires N forward and backward passed through the model to compute attributions ($N$ being the number of approximation steps from Eq. 3). Fortunately, approximation steps are independent of each other and we can compute them in parallel by batching them. Thus, the computational complexity scales linearly by a factor of $N/B$ respective to the complexity of the forward and backward pass (As $B$ is a constant the complexity is $O(N)$). Attributions to intermediate layers do not require the computation of the full backward pass, and are thus cheaper. We will include this information into the revised version of the paper.

---

### Official Review · Reviewer_4954 · 2023-08-02

**Soundness:** 4

**Excitement:**

3: Ambivalent: It has merits (e.g., it reports state-of-the-art results, the idea is nice), but there are key weaknesses (e.g., it describes incremental work), and it can significantly benefit from another round of revision. However, I won't object to accepting it if my co-reviewers champion it.

**Paper Topic And Main Contributions:**

The paper proposes a method to measure feature attributions for siamese encoders. The idea is to break out the compatible output score into several components related to input feature pairs, using path integrals proposed by Sundararajan et al 2017.

**Reasons To Accept:**

The paper is an interesting extension to Sundararajan et al. 2017 for siamese encoders (pair-wise attributions). The paper not only shows why the idea works in theory, but also demonstrate how to apply that in practice using approximation.
* In terms of theory, it is nice to see how reference input r is chosen.
* In terms of practice, the paper fairly shows the approximation ability, and that rightly points out the weaknesses of the choices related to input r (table 1).



**Reasons To Reject:**

Although being able to contribute to explainable AI, the paper seems to have narrow applications due to siamese encoders. Could the approach be applied to a wider range of network architectures, e.g. for machine translation?

The paper doesn't provide evidence whether the computed pair-wise feature attributions faithful or not.

**Reproducibility:**

3: Could reproduce the results with some difficulty. The settings of parameters are underspecified or subjectively determined; the training/evaluation data are not widely available.

**Reviewer Confidence:**

2: Willing to defend my evaluation, but it is fairly likely that I missed some details, didn't understand some central points, or can't be sure about the novelty of the work.

---

> ### Author Rebuttal · Authors · 2023-08-28
>
> Thank you for your comments. We respond to the two most substantial aspects.
>
> ### Applicability to other architectures
> Regarding your question whether our method could be applied to architectures other than Siamese encoders which deal with multiple inputs, we admit that our current study focuses on Siamese encoders since they are widely used and arguably one of the least interpretable type of model.
> Now, our Ansatz from Eq. 7 is entirely general to any differentiable model with two inputs. In the following, our derivation, however, is specific to the Siamese architecture – the separation of the two integrals in Eq. 9 is ultimately what allows our attributions to take the comparatively simple form of a matrix. Other architectures call for a more general treatment, or will result in another architecture-specific solution.
>
> ### Faithfulness
> You are right that we actually do not explicitly comment on faithfulness, which we should.
> We refer to attributions being „provably correct“ (line 127-129) because Eq. 2 is an equality. Thus, our attributions faithfully explain which features (or tokens) of the inputs the model attends to, in order to make predictions. This is different from e.g. attention visualization as it does not account for the full computational graph of the model. In lines 271-273 we conclude that our attributions „can explain which feature-pairs are relevant to individual predictions, but not why“. This is because our method *cannot* explain the reasoning within the model — not faithfully and not at all.

---

### Official Review · Reviewer_PsQ5 · 2023-08-04

**Soundness:** 3

**Excitement:**

3: Ambivalent: It has merits (e.g., it reports state-of-the-art results, the idea is nice), but there are key weaknesses (e.g., it describes incremental work), and it can significantly benefit from another round of revision. However, I won't object to accepting it if my co-reviewers champion it.

**Paper Topic And Main Contributions:**

This paper presents an attribution method, in other words a method to study the relation between the global score of a neural network and the input (or intermediate) features, specially tailored for Siamese networks based on the method of integrated gradients.
To accommodate with the design of Siamese networks, and to attribute scores to feature interactions across the two input sentences, this method is extended to deal with 'integrated jacobians', and these interactions can then be summarized in a matrix form.
These feature interactions can be factorized further by summing over the features corresponding to the same token in order to give token-to-token attributions.
This method relies on the computation of integrals, which are approximated as sums over (the differential of) the objective applied to points close to each other,
the quality of the approximation depending on the number of intermediate points.


The rest of the paper consists of experiments on the STS-benchmark. Different systems are fine-tuned (MPNet, Roberta, distillRoberta) with the proposed attribution method.

First the new objective is tested, showing it can be used in practice with models that have been already trained as sentence encoder. In other words, vanilla pretrained transformer cannot be directly trained and achieve good results.
Although no hypothesis is fathomed by the authors, I suspect that the change in the objective where the score is now unbounded (above and below) could lead to unstable gradient during training.
This would justify the use of this method as a fine-tuning procedure, where the search space the gradient wrt the loss function are small.

Then the quality of the approximation is tested, and this is where I was disappointed by the experimental results. It seems that the approximation gap of the integrated jacobian must be really tight, and so this method is unable in practice to use the input features as the abstract and the introduction would let the reader thinks. In practice the attribution method can only be used with the highest layers, or must be reserved to shallow networks.

The distribution of the attributions is then studied on token-token pairs directly and also as POS-POS pairs. In each case, a few attributions are responsible for the majority of the final scores but in order to get reliable results, the vast majority of attributions must be taken into account.
This might be an avenue for future research: can the objective be modified to encourage sparsity?

Overall, the contribution looks promising but:
- The experiments are not properly described. For instance neither the models nor the corpora used in the experiments are presented. Moreover, what the Figures or Tables represent are not really explained, and the conclusion are directly given. This all adds up to make readers struggle to fully understand what they are presented with.
  - For instance, Table 1 is really difficult to understand, because we do not know precisely what the "labels" are, nor what the "unmodified models with identical training" stand for. The accompanying test (l.163-l.178) does not help here: which ones are the "vanilla transformers", what is a "Siamese setting"? These expressions could be applied to numerous objects, and in this context it is too vague.
  - In Section 3.1, l. 185-186, the "error between the sum of attributions and the score as a function of N". This is ambiguous since "as a function of N" applies to "the error" and not to "the score". In Figure 2, the legend covers the left part of the graph
  - In Section 3.2, Figure 3 is too vaguely described (l.208-212): authors need to describe the axes, the plot etc. the cognitive burden adds up again. I fail to understand the meaning of the example in Appendix C (which should be in the main text) to address matches/mismatches. Again, it may be because the lack of explanation
- I had to read the appendix (A and B) carefully, and the paper is not intelligible otherwise, so to me this material should be in the main paper.
- The paper refers several times to figures in Appendix, for instance Fig.6 in appendix C in Section 3.2, and they should be incorporated in the main paper.
- the authors should provide more details about the training setup. Appendix H does not give enough details to reproduce the results (I am not talking about running the code in the supplementary zip file but being able to redo the experiments from the description given in the paper) **EDIT: see discussion, this is irrelevant**
- I found the first paragraph of the conclusion a bit puzzling. It seems to imply that computing attribution in this setting is less compute intensive than with other methods, especially for deeper layers, but no comparison is performed.

**Reasons To Accept:**

- The method is promising, and shows how the integrated gradient method is adaptable to different settings.

**Reasons To Reject:**

See above for a list of my concerns. To sum up, this paper should have been a long paper. As it is, it lacks clarity, and uses appendices to compensate for its succinctness.

**Reproducibility:**

4: Could mostly reproduce the results, but there may be some variation because of sample variance or minor variations in their interpretation of the protocol or method.

**Reviewer Confidence:**

4: Quite sure. I tried to check the important points carefully. It's unlikely, though conceivable, that I missed something that should affect my ratings.

**Typos Grammar Style And Presentation Improvements:**

I believe there is an error in equation 7 (appendix b): $f(a,y) - f(a,y)$ should be $f(a,y) - f(r,y)$.

---

> ### Author Rebuttal · Authors · 2023-08-28
>
> Thank you for the detailed feedback and your deep engagement with our study. In the following, we reply to your most substantial points of criticism.
>
> ### Derivation in the Appendix
> Presenting a main result in the main body of a paper and relegating the formal derivation/proof to an appendix has become a relatively standard presentation method (compare Hu et al. NAACL 2022, Mou et al. ACL 2023, etc. etc.). In this way, we can provide the application-oriented reader with a usable model, and the technically oriented readers may consult the appendix for a deeper understanding. If the paper is accepted, we will integrate the most important parts of the derivation in the main body. We would still argue that our size of study is appropriate for a short paper.
> Regarding the typo in Eq. 7, you are absolutely right. The first input to the second term must, of course, be the reference $r$ and not the input $a$. We will fix it.
>
> ### Clarity, Detail, Reproducibility
> Regarding your criticism about lack of clarity, we will take them into account when revising the paper. In particular, we acknowledge the suboptimal presentation in Tab 1.
> We disagree about the lack of detail, though: The STS benchmark is one of the most widely used and known datasets in semantic benchmarking and, in our opinion, does not require an explicit introduction. Similarly, the models we use are standard Huggingface downloads (we will add the specific links).
> We also do not agree with your comments regarding lack of reproducibility: We have submitted the full code of our implementation and provide all relevant hyper-parameter settings in Appendix H. We would be grateful to know what specific information you feel is missing, and add it.
> Admittedly, the implementation of our method is not trivial. We could add a paragraph about central implementation concepts in order to close the gap between theory and code.
>
> ### Vanilla transformers
> It is a misunderstanding that our method does not achieve good results on vanilla transformers that have only been pre-trained on language modeling. They do not show competitive predictive performance for semantic similarity, but this is entirely expected as they were never trained on semantic similarity before. They also do not adapt to the change of objective as well (Roberta being an extreme case). But this instability is also very likely due to the lack of large scale pre-training for semantic similarity.
> These models, however, **do have excellent attribution accuracy**. This is described in Section 3.1 and Appendix D.
> In this work they are really of secondary interest, because they are not trained to predict similarities. We mostly include them to evaluate attribution accuracies on a wider range of models. We will make this point clearer in a revised version.
>
> ### Gradient Stability
> It is a reasonable hypothesis that gradients may become unstable due to the fact that prediction scores are not naturally bound to [0, 1] by the dot-product. However, in practice we do not experience any issues in this regard. Training runs converge smoothly, and the model adjusts its predictions to the required range well. We believe the drop in predictive performance compared to a cosine objective is largely due to the fact that a dot-product does not preserve self-similarity (as discussed in the limitations section).
>
> ### Applicability to shallow layers
> You conclude that the method can only be used with the highest layers or in shallow networks. Generally it is true that earlier layers require more approximation steps – however, this is not a *conceptual* problem but a *practical* problem of approximation. Theoretically, the attribution error is guaranteed to converge to zero for sufficiently large $N$, because the sum in Eq. 3 will converge against the integral (by definition of the Riemann integral). In Fig. 2 we present the attribution error of the S-MPNet model, which has the best predictive performance (cf. Table 1), but does not yet have good attribution accuracy for shallow layers with $N<2500$. This may have been misleading; Fig 7 shows that other models converge much more readily.
> The only reason why we cannot run our method with a higher number of approximation steps $N$ is lack of computational resources. However, our code does support arbitrary large $N$ and could be run on a more powerful setup to achieve better attribution accuracy.
> Finally, we would like to emphasize that attributions to intermediate layers do already provide meaningful insights and will be a good compromise to run large scale analyses.
>
> ### Conclusion
> We apologize for the apparent misunderstandings arising from the first paragraph of the conclusions. First, we never claim that our method is computationally less intensive than other methods. We only claim that its attributions are provably correct (line 127-129, cf. Faithfulness - reviewer 4954). Second, to the very best of our knowledge, there exist no other interpretability methods for bi-encoder models. Standard integrated gradients, relevance backpropagation, or attention visualization may provide feature importance attributions to individual inputs. However, no other method can explain which combination of aspects of the two inputs contributed to their similarity. For this reason, comparison against other methods is difficult. We hope to make these points clearer in a revised version of the manuscript.
>
> ### Future Work - Sparsity of Attributions
> You wonder whether the objective could be modified to encourage sparsity. This is definitely an interesting point. In fact, we were somewhat surprised by how sparse attributions already are (cf. Fig. 4 - few attributions often explain a large fraction of the prediction).
> A priori, the main advantage of our method is that it is provably *faithful* to the behavior of the model, which is at odds with the idea of using it to impose an additional constraint on the model during training – such as sparsity.
> That being said, it could be possible to use output layer attributions for this regard, which are however the least informative. Utilizing attributions to intermediate layers would be tricky as they require backpropagation themselves.

---

### Meta-Review · Area_Chair_8naN · 2023-09-17

**Recommendation:** 4

**Metareview:**

The paper adapts Integrated Gradients to siamese encoders (which can be useful e.g. in tasks that learn similarity judgements for two texts). For this, the authors introduce “integrated jacobians” that summarize feature interactions between the two inputs in a matrix form.

The reviewers agreed that the method and its theoretical justification are solid. They also appreciated that the authors showed how it can be used in practice and discussed the potential weaknesses (e.g., the choice of the baseline input r, etc.) There were concerns about applicability of the method to early layers and other architectures, but those were resolved during discussion. Overall, the reviewers are positive about the paper.

---

### Decision · Program_Chairs · 2023-10-07

**Decision:**

Accept-Main

**Comment:**

The paper adapts Integrated Gradients to siamese encoders (which can be useful e.g. in tasks that learn similarity judgements for two texts). For this, the authors introduce “integrated jacobians” that summarize feature interactions between the two inputs in a matrix form.

The reviewers agreed that the method and its theoretical justification are solid. They also appreciated that the authors showed how it can be used in practice and discussed the potential weaknesses (e.g., the choice of the baseline input r, etc.) There were concerns about applicability of the method to early layers and other architectures, but those were resolved during discussion. Overall, the reviewers are positive about the paper.